# An improved diagnostic method for taurodontism and a comparative study on its effectiveness evaluation

Yunmeng Da[1]*, Le Zhang[2], Zhihong Chai[1], Hongfang Du[1], Lele Hao[1], Li Zhang[1], Zhiyin Zhang[1], Yongfan Shen[1]*

1 Department of Prothodontics, Hebei Eye Hosipital, Hebei Provincial Key Laboratory of Ophthalmology, Hebei Clinical Medical Research Center of Eye Diseases, Xingtai, Hebei, China, 2 Department of Oral and Maxillofacial Surgery, Hebei Eye Hosipital, Hebei Provincial Key Laboratory of Ophthalmology, Hebei Clinical Medical Research Center of Eye Diseases, Xingtai, Hebei, China

* dayunmeng@126.com (YD); syf200506@163.com (YS)

**Data Availability Statement:** All relevant data are within the manuscript and its Supporting information files.

## Abstract

### Objective

The two commonly used diagnostic methods for taurodontism are susceptible to aging changes, mastication wear and other factors. Therefore, this study proposed an improved diagnostic method for taurodontism, and compared it with the previous two methods as a supplement for taurodontism diagnosis.

### Methods

The included patients were aged 10–89 years and admitted to the Department of Stomatology of Hebei Eye Hospital from June 1, 2022 to May 31, 2023. Eighty cone-beam computed tomography images were divided equally into 4 groups: 10–29, 30–49, 50–69, and 70–89 years old. The right mandibular first molars were selected as measurement objects. Firstly, | BD| and taurodontism index (TI)-related parameters were measured using Shifman and Chanannel's method and crown-body(CB) and root (R) lengths was measured by Seow and Lai's method. The improved method used the length from the cementoenamel junction (CEJ) to the root bifurcation point(body, B)and the root length(root, R)as the measurement objects. Finally, TI, CB/R ratios, and B/R ratios were calculated according to the formulas given below. One-way ANOVA analysis was mainly used to compare the differences in the values, indices and ratios of taurodontism among different age groups ($p < 0.05$).

### Results

With the increase of age, |BD| and TI values decreased significantly ($p < 0.01$). The CB/R ratios of 70–89 years group were significantly lower than those of the other three groups ($p < 0.01$). Ratios derived from the improved method were significantly lower in the 70–89 years than in 10–29 years group ($p < 0.05$).

**Funding:** Xingtai Science and Technology Project, NO.: 2022zz081, Yun-meng Da. The funders had no role in study design, data collection and analysis, decision to publish, or preparation of the manuscript.

**Competing interests:** The authors have declared that no competing interests exist.

## Conclusions

The |BD| and TI parameters proposed by Shifman and channel are significantly influenced by age. The measurements of Seow and Lai (CB/R ratios) were less affected by age compared with those of the former. The improved method(B/R ratios) was least affected by age, which would reduce error and bias in the measurement of taurodontism and obtain more objective results in older patients.

## Introduction

In 1913, Keith proposed the concept of "taurodont," characterized by a shortened tooth root and an enlarged pulp chamber, which is common in ungulates (such as cows) [1], then it was classified into hypotaurodontism, mesotaurodontism, hypertaurodontism by Shaw et al. [2]. In 1978, taurodontism was diagnosed by measuring the |BD| values (the vertical distance from the CEJ to the highest point at the bottom of the pulp chamber) and taurodontism index (TI) on oral radiograph [3]. Approximately a decade later, Seow and Lai used the crown-body/root (CB/R) ratio as an indicator of taurodontism [4]. These two methods have been used in most studies [5–8].

In previous studies, the prevalence of taurodontism varied greatly among races and countries. The lowest incidence of taurodontism is 0.1% in western Saudi Arabia population [9], whereas the highest is 52.4% in the Chinese population [6]. Previous research by our group found that the improper selection of diagnostic methods for taurodontism was the main reason for the high and low incidence of taurodontism in different countries or populations. The two commonly used measurement methods of taurodontism was affected by aging changes, mastication wear, and other factors, leading to biases in the measurement results. In this study, to provide a more accurate measurement method and standard for the diagnosis of taurodontism, which is less affected by age, we improved the CB/R ratio method and used the CEJ point to replace the lowest point of the dental crown, and compared it with the previous two to evaluate its effectiveness.

## Materials and methods

### 1. Subjects

This study was approved by the Medical Ethics Committee of Hebei Eye Hospital in China (Nos.: 2019KY002 and 2022KY14). The consent was not obtained, because only the cone-beam computed tomography (CBCT) images were collected, and the data were analyzed anonymously. Only the authors could access information that identified individual participants during or after data collection. The patients were admitted to the Department of Stomatology of Hebei Eye Hospital from June 1, 2022 to May 31, 2023. Then, the data were accessed for research purposes, starting on June 1, 2023, which took approximately 2 weeks to complete. The inclusion criteria were mandibular first molars with fully developed roots, and no obvious caries or tooth defects affecting the measurement. We collected 80 CBCT images from patients randomly aged between 10 and 89 years and divided them evenly into 4 groups (20 images per group): 10–29 years, 30–49 years, 50–69 years, and 70–89 years, with 20 images in each group. The right mandibular first molar was selected as the measurement object.

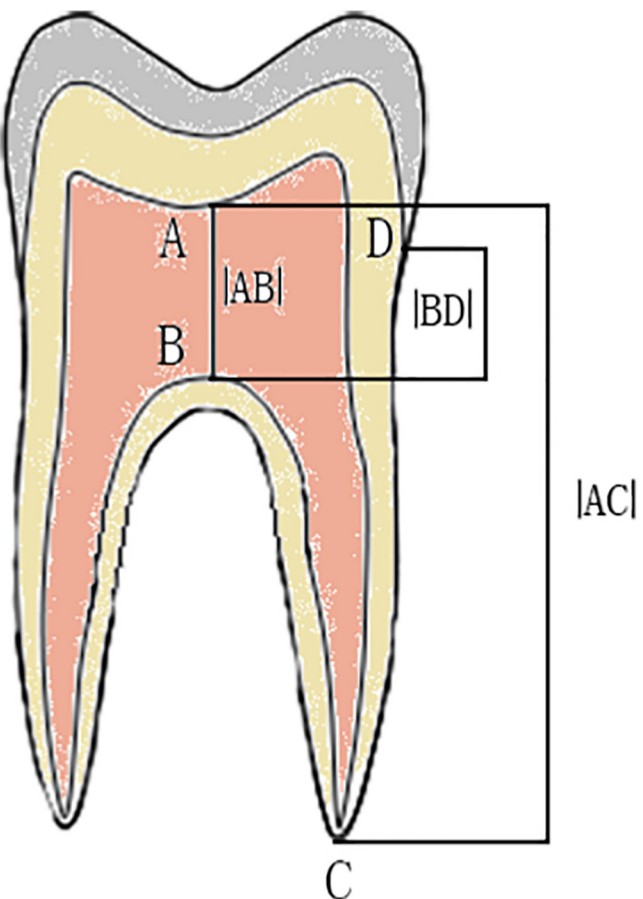

**Fig 1. The measurements of Shifman and Chanannel: |BD| is the vertical distance from the CEJ to the highest point at the bottom of the pulp chamber.** |AB| is the vertical distance between the lowest point at the top of the pulp chamber and the highest point at the bottom of the pulp chamber; |AC| is the perpendicular distance between the lowest point at the top of the pulp chamber and the apex of the longest root. CEJ, cementoenamel junction.

## 2. Diagnosis criteria

First, 80 teeth were measured using the method proposed by Shifman and Chanannel to obtain |BD| and TI values. Method: |BD| was the vertical distance from the CEJ to the highest point on the pulp chamber floor [3]. TI = |AB|/|AC|×100, whereas |AB| was the vertical distance from the lowest point of the pulp chamber ceiling to the highest point of the plasma chamber floor and |AC| was the vertical distance from the lowest point of the ceiling to the longest root tip (Fig 1).

Subsequently, the 80 teeth were measured again using the method described by Seow and Lai. CB refers to the vertical distance from the lowest point of the occlusion surface to the root bifurcation, and R refers to the vertical distance from the root bifurcation to the longest root tip. The CB and R values related to the CB/R ratio were obtained, and CB was divided by R to get CB/R ratios. (Fig 2) [4].

Eighty teeth were evaluated using the improved method. B refers to the vertical distance from the CEJ to the root bifurcation, and R refers to the vertical distance from the root

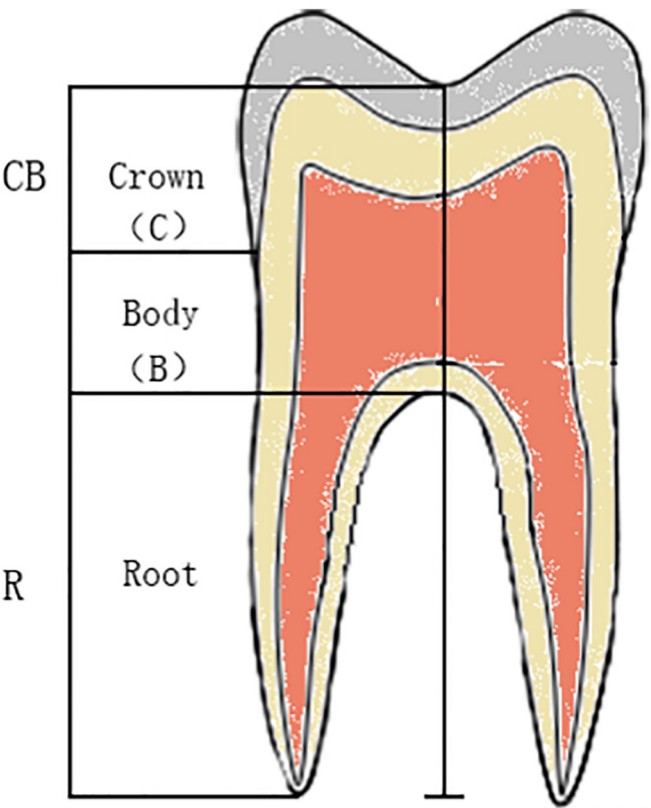

**Fig 2. The measurements of CB/R ratio and B/R ratio: Crown (C) is the vertical distance from the lowest point of occlusion surface to the CEJ.** Body (B) is the vertical distance from the CEJ to the Root bifurcation point. and Root (R) is the vertical distance from the root bifurcation point to the longest root tip. CEJ, cementoenamel junction.

bifurcation to the longest root tip. Similar to the CB/R ratio, B and R values were related to B/R ratio, and B was divided by R to obtain B/R ratios (Fig 2).

### 3. Analysis and statistics

Firstly, the values of |BD|, |AB|, |AC|, CB, B and R were measured by CBCT using Image Pro plus 6.0 software. The index was calculated using the following formula. SPSS16.0 software analysis data, with P<0.05 considered statistically significant. The homogeneity test of variance showed that the levels of variance in the TI were heterogeneous using Levene;s test(P<0.05), and the Kruskal-Wallis H test and Mann-Whitney U tests were used to analyze these data. Other data conformed to a normal distribution(P>0.05), and statistical analyses were performed by one-way ANOVA and LSD test.

## Results

With increasing age, |BD| values decreased significantly (p<0.01) (Fig 3). The TI also decreased significantly with age (p< 0.01) (Fig 4). In the first three age groups, the CB/R ratios tended to decrease but were not significantly different (p> 0.05), until the 70–89 age group showed a significant decline compared to the other groups(p< 0.01) (Fig 5). The B/R ratio also showed a downward trend, but only the 70–89 years group showed a significant decline compared with the 10–29 group(p< 0.05) (Fig 6). The data are presented in Table 1.

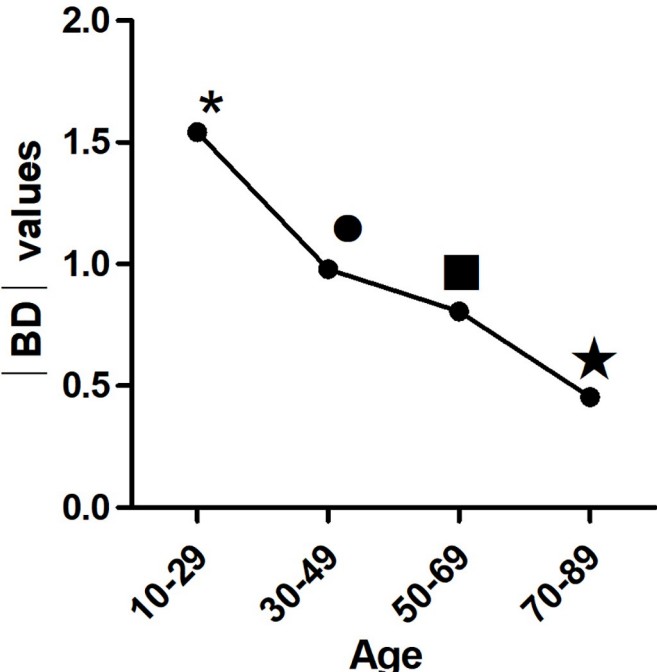

**Fig 3. The variation trend of |BD| value with age: With the increase of age, the |BD| values decreased significantly ($p < 0.01$).** * $p < 0.01$ vs other groups. ● $p < 0.01$ vs other groups except 50–69 years group($p > 0.05$). ■ $p < 0.01$ vs other groups except 30–49 years group($p > 0.05$). ★ $p < 0.01$ vs other groups.

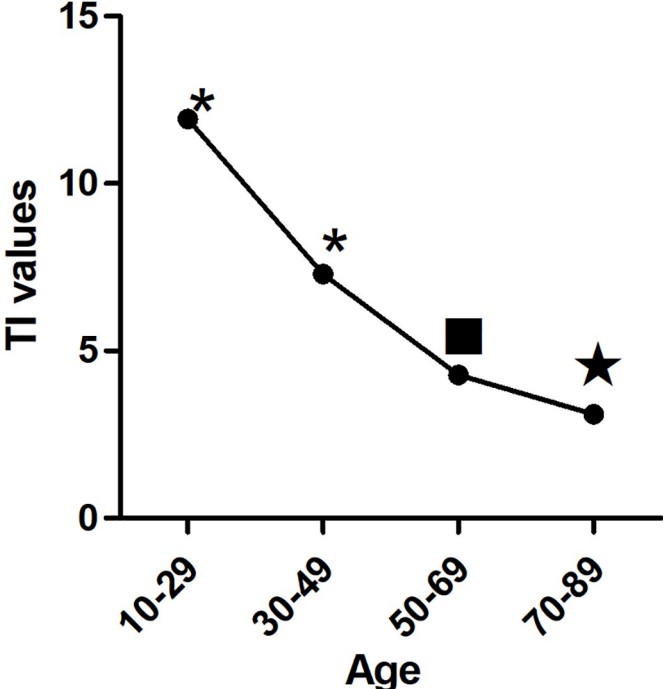

**Fig 4. Trend of TI values with age: The TI values decreased significantly with the increase of age,** * $p < 0.01$ vs other groups. ■ $p < 0.01$ vs other groups except 70–89 years groups($p > 0.05$). ★ $p < 0.01$ vs other groups except 50–69 years groups($p > 0.05$).

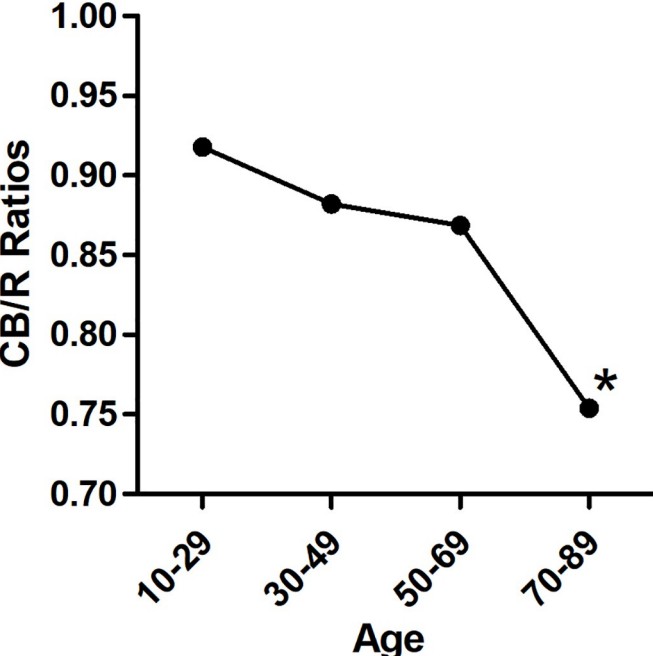

**Fig 5. Trend of CB/R ratios with years: There were no significant differences in CB/R value between the first three groups(p>0.05).** The 70–89 years group showed significant decrease vs the other groups significantly. * p<0.01 vs the other groups.

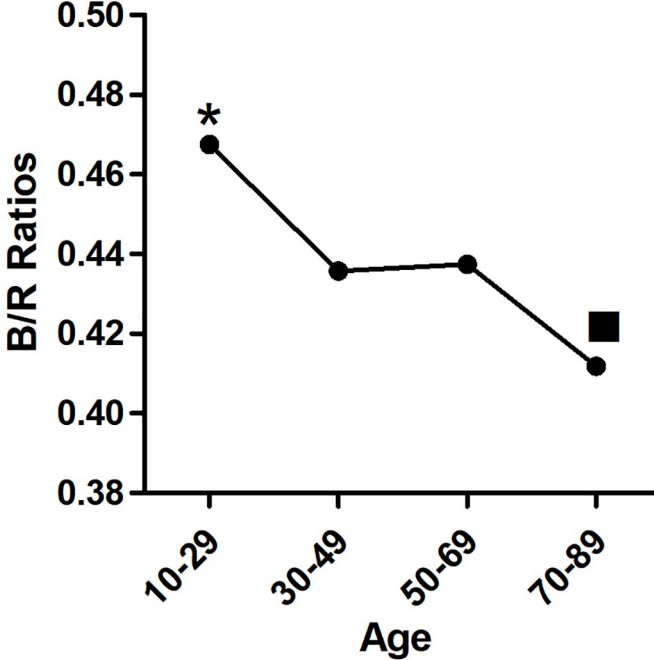

**Fig 6. Trend of B/R ratios with years: There were no significant differences in CB/R values between the first three groups(p>0.05).** The 70–89 years group showed significant decrease vs. the 10–29 years groups. * p<0.01 vs. the other groups.

**Table 1. The |BD|, TI, CB/R and B/R values of the mandibular first molar were (X ± s).**

| methods | 10–29 years | 30–49 years | 50–69 years | 70–89 years |
|---|---|---|---|---|
| |BD| | 1.54±0.43 | 0.98±0.38 | 0.81±0.31 | 0.45±0.23 |
| TI | 11.93±2.69 | 7.29±3.56 | 4.27±2.37 | 3.10±1.24 |
| CB/R | 0.92±0.11 | 0.88±0.09 | 0.87±0.09 | 0.75±0.08 |
| B/R | 0.47±0.07 | 0.436±0.076 | 0.437±0.090 | 0.41±0.07 |

## Classification of the improved method

Since the modified method was an improvement on the basis of CB/R ratio, we still used the grade conversion method of CB/R ratio, normal (CB/R≤1.10), hypotaurodontism (1.10<CB/R≤1.30), mesotaurodontism (1.30<CB/R≤2.0), and hypertaurodontism (2.0<CB/R) [4]. It was estimated that the CB/R ratio was estimated to be approximately twice that of B/R, which we conclude had the following values: normal (B/R≤0.55), hypotaurodontism (0.55<B/R≤0.65), mesotaurodontism (0.65<B/R≤1.0), and hypertaurodontism (1.0 < B/R).

## Discussion

Through the analysis of the research results, it was found that the values of |BD| and TI using Shifman and Chanannel's were significantly affected by age and showed a sharp downward trend which would undoubtedly lead to bias in the measurement of taurodontism, and false-negative results would occur in the measurement of older patients. The results were easy to explain, with increasing age, the top point of the pulp chamber was down growth, and the bottom point was up growth, resulting in a reduction in the overall height of the pulp cavity [10–12]. |BD| was the distance from the CEJ to the pulp chamber floor. Although the CEJ position did not change, an,increase in the pulp chamber floor lead to the decrease of |BD|. The TI was related to the height of the top and bottom of the pulp chamber; therefore, it was had been significantly reduced with aging. Many studies on the incidence of taurodontism have adopted the Shifman and chanannel method, some of which selected adolescents as research objects and obtained relatively accurate results [5, 13], while others targeted at different age groups, including middle-aged and elderly people [7, 14]. It was obviously inappropriate to adopt this method, which would lead to an underestimation of the incidence. Obviously, taurodontism does not disappear with age.

The measurements of Seow and Lai (CB/R ratio) were less affected by age than those of the Shifman and Chanannel method. This was primarily because the lowest point of the occlusion surface was selected using this method. Although the occlusion surface also decreases with chewing wear, the process would takes decades to reach its lowest point. After 70 years of age, the CB/R ratio decreased significantly because the enamel of the occlusion surface was worn out, softer dentin was exposed, and the wear rate accelerated. However, it was not ruled out that some patients with acidosis or gastroesophageal reflux disease would experience severe wear in advance due to the acid erosion of dental minerals, resulting in low measurement values also.

The improved method proposed in this study uses the CEJ points as markers. On the one hand, the CEJ point was not affected by age-related changes after formation; on the other hand, CEJ points are generally located below the gums, and factors such as wear and acidosis rarely affect them, therefore the diagnosis of taurodontism based on this was more accurate and objective. We observed that compared with the first two methods, the improved method was the least affected by age. Only the age group 70–89 and 10–29 years old showed a

significant decline. Similar to the results of Pach et al., the incidence of human taurodontism varied at different periods [15]. We believe that this might be related to the different dietary habits and food types of the two generations. The growth and development period of the elderly was the transition between the old and new societies in China, when food was mainly coarse grains and the mastication force was strong. The stimulation of tooth root development was longer and the root bifurcation was larger, leading to a reduction in the B/R ratios. The growth and development period of 10–29 years old was in the modern era with abundant fine soft food, the mastication force was weaker than that of the 70–89 years group, shorter tooth root development, and the ratio correspondingly enhanced. A study found that the cementum in the apical area increased with age, and the root length increased slightly when the tooth was older [16], which may also be a reason for the decline in the index of the 70–89 years group.

Current studies on taurodontism has mainly focused on its association with systemic diseases and related pathogenic genes. Related systemic diseases such as tricho-dento-osseous (TDO) syndrome has also been reported [17–20]. Related pathogenic genes include: NFI-C/CTF, EDA, RUNX2,WNT10A, WNT10B, GREM2, etc. [21–26]. There have been no discussions on the measurement of taurodontism. Two statistical studies on the incidence of taurodontism in the Chinese population produced very different results. Da et al. found that the incidence of taurodontism was as high as 52.4% [6]; however, Li et al. calculated the incidence of taurodontism in premolars and molars was only 29.14%, and both study populations were approximately 30 years old [7]. The two methods used to calculate the taurodontism index were different. Therefore, the taurodontism index methods might be biased by age. The volume of the pulp chamber decreases with age, and the accompanying changes in the position of the top and bottom of the pulp chamber inevitably affect the measurement results of the Shifman and Chanannel method [1, 10]. Tooth abrasion with age certainly affects the crown-to-root ratio method but will appear later [27]. Once the CEJ is formed, it may be worn by tooth brushing or by acidic substances on the labial or buccal side [28, 29], but the position of the CEJ in the proximal and distal regions does not generally change with age. Therefore, the improved method is less affected by age.

Taurodontism poses challenges to the treatment of various dental condition, including endodontic diseases, periodontal diseases, orthodontics and prosthetics [8]; Therefore, it is extremely important to obtain accurate incidence rates. To sum up, the selection of diagnostic methods for taurodontism should be based on different age groups to choose different detection methods. The Shifman and Chanannel method can be used to measure juveniles with fully developed root tips [3]. For middle-aged patients, Seow and Lai's method was selected as much as possible [4]. This improved method is more practical for elderly patients, with severe masticatory wear or obvious depression of the occlusion surface. This study proposes an improved diagnostic method for taurodontism and compares its effectiveness and provides a reference for clinicians to accurately diagnose and treat taurodontism in clinical practice.

## Supporting information

**S1 File. Data for Table 1 and Figs 3–6.** The first column is age. The 2–4 column is |BD|, TI, CB/R values, which are data sources in Figs 3–5 respectively. The last column is the values derived from the improved method, data for Fig 5.
(XLSX)

## Author Contributions

**Conceptualization:** Yunmeng Da, Yongfan Shen.

**Data curation:** Yunmeng Da, Le Zhang.

**Formal analysis:** Yunmeng Da, Le Zhang.

**Funding acquisition:** Yunmeng Da.

**Investigation:** Yunmeng Da, Zhihong Chai.

**Methodology:** Yunmeng Da, Zhihong Chai.

**Project administration:** Yunmeng Da, Hongfang Du, Lele Hao, Li Zhang.

**Resources:** Hongfang Du, Lele Hao, Li Zhang.

**Supervision:** Zhiyin Zhang.

**Validation:** Zhiyin Zhang.

**Writing – original draft:** Yunmeng Da.

**Writing – review & editing:** Yongfan Shen.

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
