## [Decision Letter · Decision Letter 0]

26 Jan 2024

PONE-D-23-41512An improved diagnostic method for taurodontism and a comparative study on its effectiveness evaluationPLOS ONE

Dear Dr. Da,

Thank you for submitting your manuscript to PLOS ONE. After careful consideration, we feel that it has merit but does not fully meet PLOS ONE’s publication criteria as it currently stands. Therefore, we invite you to submit a revised version of the manuscript that addresses the points raised during the review process.

We look forward to receiving your revised manuscript.

Kind regards,

Ranjdar Mahmood Talabani

Academic Editor

PLOS ONE

Journal Requirements:

The name of the colleague or the details of the professional service that edited your manuscript.A copy of your manuscript showing your changes by either highlighting them or using track changes (uploaded as a *supporting information* file).A clean copy of the edited manuscript (uploaded as the new *manuscript* file).

3. Please be informed that funding information should not appear in the Acknowledgments section or other areas of your manuscript. We will only publish funding information present in the Funding Statement section of the online submission form. Please remove any funding-related text from the manuscript. 

"Xingtai Science and Technology Project, 2022zz081, Yun-meng Da"

**Additional Editor Comments:**

Greetings,

The author(s) should follow instruction suggested by peer reviewer.

Reviewers' comments:

Reviewer's Responses to Questions

**Comments to the Author**

1. Is the manuscript technically sound, and do the data support the conclusions?

Reviewer #1: Yes

2. Has the statistical analysis been performed appropriately and rigorously? 

Reviewer #1: Yes

3. Have the authors made all data underlying the findings in their manuscript fully available?

Reviewer #1: Yes

4. Is the manuscript presented in an intelligible fashion and written in standard English?

Reviewer #1: Yes

5. Review Comments to the Author

Reviewer #1: Minor corrections should be made.

The manuscript needs minor grammar checking with proper punctuation.

The authors should make the required corrections.

1-Abstract

The abstract is well-organized and clear.

The authors should make the required corrections that are highlighted.

2- Introduction

1. The authors should make the required corrections that are highlighted.

2. Give a definition of taurodontism and add refs to the introduction.

3-Material and method

1. The author should make the required corrections that are highlighted.

2. The author should mention the country of study to compare the study's findings with those of other countries in the discussion.

4- Results

1. The author should make the required corrections that are highlighted.

2. The author should change Fig 6, as a slight increase in B/R ratio was seen in the 50-69 age group compared to 30 -49, as shown in Fig 6.

3. In a classification of the improved method, correct symbols.

4. In a classification of the improved method, the authors should mention their results of CB/R and B/R according to this grade.

5- Discussion

1. The author should make the required corrections that are highlighted.

2. The author should put refs for other studies that used Shifman & channel method to determine taurodontism.

3. The authors should discuss their research findings regarding the (grade) of taurodontism of improved classification B/R , compare results to other countries, explain the cause of the difference, and add refs.

4. The author should add other references to the discussion (The Shifman & channel method can be used to measure juveniles with fully developed root tips. For middle-aged patients, Seow and Lai’s method was chosen as far as possible)

6- References

1. The author should make the required corrections that are highlighted.

2. Add other references, as the number of refs is few.

6. PLOS authors have the option to publish the peer review history of their article (what does this mean?). If published, this will include your full peer review and any attached files.

Reviewer #1: No

---

## [Author Response · Author response to Decision Letter 0]

7 Apr 2024

Dear reviewer：

Thank you for your careful consideration and pertinent comments. According to the requirements, we have carefully revised the manuscript, and implemented one modification and response to the revision suggestions in the rosponds to reviewer letter. As requested, we have included the funding information below.

Yours

Sincerely

Yunmeng Da

---

## [Decision Letter · Decision Letter 1]

15 Apr 2024

An improved diagnostic method for taurodontism and a comparative study on its effectiveness evaluation

PONE-D-23-41512R1

Dear Dr. Yunmeng Da

We’re pleased to inform you that your manuscript has been judged scientifically suitable for publication and will be formally accepted for publication once it meets all outstanding technical requirements.

Kind regards,

Ranjdar Mahmood Talabani

Academic Editor

PLOS ONE

Additional Editor Comments (optional):

Reviewers' comments:

Reviewer's Responses to Questions

**Comments to the Author**

1. If the authors have adequately addressed your comments raised in a previous round of review and you feel that this manuscript is now acceptable for publication, you may indicate that here to bypass the “Comments to the Author” section, enter your conflict of interest statement in the “Confidential to Editor” section, and submit your "Accept" recommendation.

Reviewer #1: All comments have been addressed

2. Is the manuscript technically sound, and do the data support the conclusions?

Reviewer #1: Yes

3. Has the statistical analysis been performed appropriately and rigorously? 

Reviewer #1: Yes

4. Have the authors made all data underlying the findings in their manuscript fully available?

Reviewer #1: Yes

5. Is the manuscript presented in an intelligible fashion and written in standard English?

Reviewer #1: Yes

6. Review Comments to the Author

Reviewer #1: (No Response)

7. PLOS authors have the option to publish the peer review history of their article (what does this mean?). If published, this will include your full peer review and any attached files.

Reviewer #1: No

---

## [Editor Report · Acceptance letter]

26 Apr 2024

PONE-D-23-41512R1 

PLOS ONE

Dear Dr. Da, 

I'm pleased to inform you that your manuscript has been deemed suitable for publication in PLOS ONE. Congratulations! Your manuscript is now being handed over to our production team.

Kind regards, 

on behalf of

Dr. Ranjdar Mahmood Talabani 

Academic Editor

PLOS ONE